# A Comprehensive Overview of the Antibiotics Approved in the Last Two Decades: Retrospects and Prospects

**DOI:** 10.3390/molecules28041762

**Published:** 2023-02-13

**Authors:** Zhenfeng Shi, Jie Zhang, Lei Tian, Liang Xin, Chengyuan Liang, Xiaodong Ren, Min Li

**Affiliations:** 1Department of Urology Surgery Center, Xinjiang Uyghur People’s Hospital, Urumqi 830002, China; jiezhang0505@aliyun.com (J.Z.); leitian115@gmail.com (L.T.); 2Faculty of Pharmacy, Shaanxi University of Science & Technology, Xi’an 710021, China; liangliangxin@aliyun.com (L.X.); chengyuanliang@gmail.com (C.L.); 3Medical College, Guizhou University, Guiyang 550025, China; xdren@gzu.edu.cn; 4College of Pharmacy, Xinjiang Medical University, Urumqi 830054, China; limin831120@163.com

**Keywords:** antibacterial resistance, superbugs, aminoglycosides, oxazolidinedione, pleuromutilin, structure–activity relationships (SARs)

## Abstract

Due to the overuse of antibiotics, bacterial resistance has markedly increased to become a global problem and a major threat to human health. Fortunately, in recent years, various new antibiotics have been developed through both improvements to traditional antibiotics and the discovery of antibiotics with novel mechanisms with the aim of addressing the decrease in the efficacy of traditional antibiotics. This manuscript reviews the antibiotics that have been approved for marketing in the last 20 years with an emphasis on the antibacterial properties, mechanisms, structure–activity relationships (SARs), and clinical safety of these antibiotics. Furthermore, the current deficiencies, opportunities for improvement, and prospects of antibiotics are thoroughly discussed to provide new insights for the design and development of safer and more potent antibiotics.

## 1. Introduction

The current problems of multidrug resistance (MDR) and emerging superbugs have led to decreased and even loss of antibiotic efficacy [1], which greatly hinders the treatment of infectious diseases; as a result, there is great demand for finding novel antibiotics that are effective and safe [2]. In 2021, the World Health Organization (WHO) reported that antibiotic resistance is a serious challenge, and by 2050, 10 million people are expected to die each year due to the frequent excessive use of antibiotics [3]. In addition, the morbidity and mortality of drug-resistant infections, which are the most feared consequences, increase annually worldwide [4].

Although antibiotic resistance continues to increase, many exciting new chemical entities have been developed as antimicrobial agents, and antibiotics can be classified into categories based on their mechanism of action: inhibition of bacterial nucleic acid replication and transcription, interference with bacterial protein synthesis, prevention of bacterial cell wall synthesis, and enhancement of bacterial cell membrane permeability [5]. It is hoped that an in-depth exploration of the relationship between the chemical structure, mechanism, and antibacterial activity of antibiotics that were marketed from 2000 to 2021 (Figure 1) will provide insights for the design and development of next-generation antibiotics.

## 2. DNA Topoisomerase Inhibitors

### Quinolones

Quinolones with 4-quinolinone scaffolds that target bacterial DNA gyrase and topoisomerase IV can form three-component complexes with enzymes and DNA and achieve antibacterial effects by inhibiting bacterial DNA helicase activity, destroying the DNA replication process, affecting mRNA and protein synthesis, and ultimately inducing bacterial death (Table 1) [6]. Quinolone antibiotics constitute the vast majority of the drugs listed in recent years (Figure 2). These antibiotics are easy to modify and have simple and stable nuclei, broad-spectrum activity, and generally acceptable safety characteristics [7].

Long-term studies have established the SARs of quinolones (Figure 3). Fluoroquinolones, which have a fluorine atom at C-6, have attracted much attention due to their potent, broad-spectrum activity, ease of synthesis, and excellent oral bioavailability potential. Specifically, the A ring of pyruvic acid is a necessary group; replacing the carboxyl group at C-3 by acidic groups (such as sulfhydryl and acetic acid) and the carbonyl group at C-4 by a thioketone group or alanyl can reduce the antibacterial activity. In addition, the antibacterial and anticancer activity of the 3-carboxylic acid scaffold can be transformed through modifications. The C-3-modified derivatives containing the N-(5-(benzylthio)-1,3,4-thiadiazol-2-yl)-carboxamide moiety are good examples of new anticancer drugs [8]. Ring B can be slightly changed compared with ring A; replacing the 1-N substituent with vinyl, fluoroethyl, cyclopropyl, or difluorophenyl can enhance the activity, and similar results can be obtained through the substitution of N by ethyl or benzene [9]. If the substituent is at C-2, steric hindrance interferes with the binding of drugs to receptors and decreases or eliminates the activity. The C-5 substituents have both potential and steric effects on the antibacterial activity from both potential and stereo factors, and better antibacterial impact may be obtained with an amino substituent. The fluorine atom boosts penetration into bacterial cells and improves the antibacterial activity to 30 times higher than that of quinolones. Furthermore, the 5-fluoroisatin moiety improves the lipophilicity of drugs while increasing the antimycobacterial activity [10]. Substituents at C-7 can directly interact with DNA helicase or topoisomerase IV, and nitrogen-containing five- or six-atom alkaline heterocycles, such as piperazine, exhibit good activity. The activity could be improved by introducing chlorine or methoxy groups in C-8 [11].

Prulifloxacin (**1**) is a lipophilic prodrug of ulifloxacin and contains a sulfur-containing four-membered ring added at C-1 and C-2 of the quinolone nucleus. To improve the oral absorption, an oxodioxolenylmethyl group was added to the 7-piperazine ring [12]. A multicenter, double-blind, randomized clinical study evaluated the efficacy and safety of prulifloxacin and levofloxacin in the treatment of respiratory tract infections and complicated urinary tract infections (cUTIs). A comprehensive comparison found that the tolerability and clinical effect of prulifloxacin and levofloxacin for respiratory tract infections and UTIs are similar [13]. Moreover, a lower frequency of administration may improve patient compliance and thus provide an advantage over ciprofloxacin [14]. The phase III clinical trial findings suggest that prulifloxacin penetrates well into target tissues and eliminates the long half-life for the treatment of lower UTIs and chronic bronchitis as well as diarrhea (NCT02439632) [15]. In short, prulifloxacin has the advantages of high bioavailability and low toxicity.

According to an SAR analysis of quinolone, the aminopyrrolidinyl group is present at C-7, a chlorine substituent is introduced at C-8, and the fluorinated cyclopropyl group is present at N-1 [16]. In addition, the chiral amine structure based on the quinolone nucleus has a cis-fluorinated cyclopropylamine group, which endows sitafloxacin hydrate (**2**) with good pharmacokinetic (PK) properties.

Besifloxacin hydrochloride (**3**) is a novel chiral fluoroquinolone that is used to treat bacterial conjunctivitis with pure enantiomers [17]. In this structure, 1-N cyclopropyl confers broad-spectrum activity against aerobic bacteria [18]. Fluorine at C-6 is an important pharmacophore for these drugs, and chlorine at C-8 enhances the antibacterial activity [19]. Compared with other fluoroquinolones, the unique combination of a 7-azepinyl ring and an 8-chloro substituent results in unique interactions with DNA gyrase and topoisomerase IV that can delay drug resistance.

Nemonoxacin (**4**) overcomes bacterial resistance to fluoroquinolones and exhibits good activity against Gram-positive bacteria (GPB). The drug has no fluorine atom at C-6, and chiral primary amine-substituted piperidine was introduced at C-7. The methoxy group at C-8 of the quinolone skeleton improves the effect on methicillin-resistant *Staphylococcus aureus* (MRSA) but reduces the efficiency against *Mycobacterium* [20]. Since the introduction of nemonoxacin, the long-established theory that C-6 fluorination enhances antibacterial ability has been contradicted, and researchers have diverted their attention to the structural modification of C-8.

In terms of structure, both pazufloxacin mesylate (**5**) and levofloxacin contain a 1-aminocyclopropyl substituent [21]. Due to the selective antagonism between the host and bacterial DNA, eukaryotic topoisomerase is not inhibited, while the supercoiled structure of bacterial DNA is destroyed by pazufloxacin. Compared with ciprofloxacin, pazufloxacin has a broad spectrum, low toxicity, high efficiency, and low sensitivity to light. Gratifyingly, pazufloxacin mesylate does not readily generate cross-resistance, and a lower incidence of side effects has been observed.

Gemifloxacin mesylate (**6**) is a pyrrolidine quinolone derivative that was approved for the treatment of community-acquired bacterial pneumonia (CABP), acute bacterial exacerbation of chronic bronchitis, and acute bacterial sinusitis caused by MDR *S. pneumoniae* [22]. In terms of its structure, an oxime-pyrrolidinyl group was inserted at C-7 to improve the lipophilicity [23]. The methoxime (CH_3_O-N=) on the side chain enhances the activity against GPB [24]. Replacement of the methyl by hydrogen or other substitutes results in a marked decrease or even loss of activity. The activity of gemifloxacin mesylate against *Streptococcus pneumoniae* (*S. pneumoniae*) is 30 times higher than that of ciprofloxacin and 4–8 times higher than that of moxifloxacin [25].

Unlike other quinolone drugs, no fluorine atom is embedded at C-6 in the quinolone skeleton of garenoxacin mesylate (**7**) [26]. Garenoxacin mesylate exerts potent effects on respiratory tract infections [27], particularly infections caused by penicillin or fluoroquinolone-resistant *S. pneumoniae* [28]. Garenoxacin mesylate effectively resolves respiratory tract infections and otolaryngology infections caused by drug-resistant GPB. Moreover, due to its broad-spectrum activity, this drug has great potential in the treatment of mixed aerobic and anaerobic infections [29].

Zabofloxacin hydrochloride (**8**) is a novel fluoroquinolone antibacterial drug that is effective against both bacterial type II and IV topoisomerases [30]. Zabofloxacin hydrochloride exhibits broad-spectrum activity and more potent antibacterial activity than ciprofloxacin, moxifloxacin and sparfloxacin against GPB, such as *Staphylococcus aureus* (*S. aureus*), *Streptococcus pyogenes* (*S. pyogenes*) and *S. pneumoniae* [31]. Although the common quinolone drugs exhibit QT interval extension, the administration of zabofloxacin hydrochloride does not significantly change the QT interval compared with the baseline values [32]. The drug is expected to be utilized in patients with cardiovascular disease combined with CABP, UTI, sepsis, bacteremia, otitis media, and endocarditis [33].

Ozenoxacin (**9**) can inhibit DNA gyrase and topoisomerase IV and exhibits high antibacterial activity against quinolone-resistant MRSA isolates (minimum inhibitory concentration (MIC) ≤ 0.0078 µg/mL) [34]. Ozenoxacin can destroy most microorganisms isolated from skin and soft tissue infections (SSTIs) and has been approved by the FDA for the treatment of pustules [35]. The global prevalence of pustules is approximately 11.2%, and the incidence is particularly high among children. Compared with other types of quinolones as well as erythromycin, gentamicin, and clindamycin, ozenoxacin is 4–16 times more effective against skin pustules [36]. Ferrer conducted a three-stage clinical trial of 1% ozenoxacin cream for the topical treatment of infectious skin diseases and found a higher clinical success rate compared with other drugs [37].

Delafloxacin meglumine (**10**) is an acidic fluoroquinolone [38] that was approved by the FDA on 1 June 2017 for the treatment of acute bacterial skin and skin structure infections (ABSSSIs) [39]. The SARs of delafloxacin are as follows: The substitution of N1 by an aromatic heterocyclic ring increases the molecular volume and surface area. The carboxyl groups at C-3 and C-4 are necessary for activity, and electron-withdrawing fluorine on C-6 and chlorine on C-8 enhance the bactericidal effect. The absence of the C-7 basic group can elevate the activity under acidic conditions [40]. Due to its low MIC against aerobic bacteria, GPB, and Gram-negative bacteria (GNB), delafloxacin provides an important supplement for the treatment of acute bacterial skin infections and a wide variety of bacterial infections, particularly drug-resistant bacterial infections.

Levonadifloxacin arginine salt (**11**, Emrok) and its prodrug alalevonadifloxacin mesylate (**12**, Emrok O) are broad-spectrum antibacterial agents that can protect against multidrug-resistant pathogens. These drugs belong to the benzoquinolizine subclass of quinolones and are administered intravenously and orally [41]. Preclinical and clinical data indicate that levonadifloxacin arginine salt and alalevonadifloxacin mesylate have better safety characteristics than vancomycin, teicoplanin, daptomycin, and linezolid [42].

Lascufloxacin hydrochloride (**13**) is an 8-methoxy fluoroquinolone that was launched in 2020 in Japan, can inhibit both wild and mutant bacterial DNA topoisomerases, and is used for the treatment of respiratory and otolaryngological infections [43].

Finafloxacin (**14**) demonstrates superior antibacterial activity at pH 5.0–6.0: its MIC under this condition is 4–8 times lower than that at neutral pH [44]. Furthermore, finafloxacin is expected to overcome melioidosis [45], which has high worldwide mortality and is difficult to cure. The pathogenic microorganism causing this disease is *Burkholderia pseudomallei* (*B. pseudomallei*), and its resistance to fluoroquinolones relies on the expression of the BpeEF-OprC efflux pump, a drug efflux pump of the resistance–nodulation–cell-division family [46]. Studies have shown that finafloxacin has the potential to treat anthrax-like diseases; however, more studies are needed to confirm this hypothesis [47].

Antofloxacin (**15**) is a new 8-amino-fluoroquinolone with significant activity against MRSA and *Staphylococcus epidermidis* [48]. The compound exhibits very little phototoxicity and has high cardiovascular safety and prominent PK properties. Furthermore, antofloxacin could inhibit CYPIA2 activity, but only slight inhibition of CYP2D-6 was observed in a study of the effect of antofloxacin hydrochloride on cytochrome P450 (CYP450) isoforms in rats (with theophylline, midazolam, chlorzoxazone, dexmedetomidine, omeprazole, and diclofenac as probes in the self-control method) [49].

Balofloxacin (**16**) has excellent antibacterial activity against both GNB and GPB [50]. Structurally, methoxy at C-8 of balofloxacin could reduce phototoxicity [51]; moreover, the low migration rate of balofloxacin into cerebrospinal fluid guarantees central nervous system safety.

## 3. Protein Synthesis Inhibitors Acting on Ribosomal Subunits

### 3.1. Tetracyclines

Tetracycline antibiotics prevent the elongation of peptide chains by interfering with the connection between the amino-tRNA and the 30S subunit of bacterial ribosomes and thus inhibit bacterial protein synthesis [52]. From the SAR perspective, the integrity of the four linear tetracycline rings dominates the antibacterial activity, whereas replacement or alkylation of the amide at C-2 reduces the activity. However, replacement of the active hydrogen on the amide group by another substituent results in no major change in antibacterial activity. Replacing or removing dimethylamine at C-4 can reduce the activity. Introducing a ketone group at C-5 or C-6 eliminates the activity. The addition of electron-withdrawing groups (e.g., F, Cl) or electron-donating groups (e.g., dimethylamino) at C-7 can improve the activity. The substituents between C-5 and C-9 are nonessential for activity, and the addition of a glycylamino group to C-9 results in enhanced antibacterial activity but lower bioavailability, as in the case of tigecycline. Furthermore, the addition of an aminomethyl group at C-9 can increase oral bioavailability, as in the case of omadacycline. Moreover, the inclusion of at least five carbons in the amine side chain increases the plasma protein binding rate. However, the hydroxyl group at C-10 and the diketone structures of C-11 and C-12 are indispensable [53]. The D ring is often modified to yield better tetracycline candidates (Figure 4) [54]. Studies conducted in recent years have shown that tetracyclines with C-7 fluorine atoms exhibit excellent PK properties and antibacterial activity, and eravacycline is a successful example that supports this viewpoint [55].

Tigecycline (**17**) is also called 9-t-butylglycylamido-minocycline (Figure 5). As the first commercialized glycylcycline antibiotic, tigecycline exerts a strong, extended-spectrum antibacterial effect due to the glycine phthalamide embedded into the minocycline skeleton [56]. This remodeling strategy also allows tigecycline to overcome the major molecular mechanisms of tetracycline resistance. Tigecycline is not affected by any efflux through tetracycline-specific efflux pumps or by the ribosomal protection mechanism of tetracycline resistance [57]. Therefore, tigecycline is an effective alternative that can address the challenges of MDR pathogens. However, due to insufficient gastrointestinal absorption, tigecycline requires intravenous administration and does not readily cross the blood–brain barrier (BBB).

Sarecycline hydrochloride (**18**) is a narrow-spectrum antibacterial drug that was first approved for the treatment of moderate to severe nonnodular acne vulgaris [58]. Sarecycline attaches N-ethyl-N,O-dimethylhydroxylamine to C-7 of the skeleton. As a result, extension of the C-7 chain could reach the ribonucleic acid channel and interact with codon A, which results in interference with tRNA regulation [59].

As a new 9-neopentylaminomethylminocycline [60], omadacycline (**19**) exerts potent effects against infections caused by MDR bacteria and shortens the patient’s hospitalization time [61]. Compared with other tetracycline drugs, omadacycline is more active against bacteria that possess efflux pumps and other drug-resistant factors [62]. Omadacycline is structurally similar to tigecycline and possesses a dimethylamino substitution at C-7. Furthermore, replacement of glycine at C-9 with an alkyl aminomethyl yields omadacycline [63]. For further optimization, extending the aminomethyl side chains to at least three carbons will improve the activity. In addition, branched alkyl chains and piperidine analogs have promising future prospects [60]. A comparison of omadacycline, minocycline, and doxycycline, administered either orally as tosylate salts or via intravenous injection, determined that omadacycline has certain advantages in therapeutic effect and safety [64].

Eravacycline (**20**), 7-fluoro-9-pyrrolidinoacetamido-6-demethyl-6-deoxytetracycline [54], exhibits improved antibacterial activity as a consequence of the introduction of the fluorine atom at C-7 and the pyrrolidine acetamide group at C-9, which suppress the protective effect of ribosomes and the efflux of the antibiotic [65]. The fluorine atom and pyrrolidine acetamide group were added to the skeleton to enhance the activity and broaden the antibacterial spectrum [66]. In a randomized double-blind trial, eravacycline had a better therapeutic effect in the treatment of complex abdominal infection than meropenem, as revealed by a cure rate of 87.5% versus 84.6% [67].

### 3.2. Aminoglycosides

An aminoglycoside is a type of glycoside antibiotic that combines an amino sugar and an aminocyclitol through a glycosidic bond. The compound has broad-spectrum activity, good water solubility, and good stability, but its clinical utilization is limited by its severe ototoxicity and nephrotoxicity [68]. Aminoglycosides target the bacterial 30S ribosomal subunit and inhibit bacterial protein synthesis. In a normal physiological environment, aminoglycosides cannot easily penetrate the cell membrane due to their low permeability, and thus, these drugs are not readily absorbed by the gastrointestinal tract and have difficulty penetrating the BBB. However, aminoglycosides have once again attracted attention since the launch of aminocyclitol (Table 2).

Plazomicin (**21**) is used for the treatment of adult patients with cUTIs, including pyelonephritis [69], and can be used to treat carbapenem-resistant Enterobacteriaceae infections and MDR Gram-negative infections. Regarding SARs, plazomicin has a 2-deoxystreptamine skeleton obtained by modifying the structure of sisomicin with hydroxyl-aminobutyric acid and introducing hydroxyethyl (Figure 6) [70]. Plazomicin prevents the action of the aminoglycoside-modifying enzyme that neutralizes sisomicin [71] and exhibits good bactericidal activity.

### 3.3. Oxazolidinones

Peptidyl transferase inhibitors prevent the synthesis of bacterial proteins by binding to the specific region of 23S RNA near the 50S subunit of the ribosomal peptide transfer center [72]. Due to their unique mechanism of action and excellent antibacterial activity, these antibiotics have been considered ideal drugs for resolving Gram-positive MDR. Oxazolidinone antibiotics are an important class of peptidyl transferase inhibitors that contain a tricyclic parent skeleton with rings A, B, and C and a key substituent at C-5 of the oxazolidinone ring (A) (Figure 7) [73]. At present, research on oxazolidinone antibacterial agents mainly focuses on the modification of the skeleton based on linezolid, including modification of the terminal morpholine ring, modification of the C-5 side chain, and construction of the central tricyclic fusion skeleton. The SAR summary shows the use of C/D-ring isosteric replacements such as tetrazolylpyridine, dihydropyridone, and 1,2,3-triazolyl-linked phenyl for the usual morpholine C-ring at the distal end of linezolid. These extended oxazolidinone antibacterial agents generally have stronger antibacterial activity and exhibit lower myelosuppressive effects and decreased monoamine oxidase inhibitory potential, especially when combined with additional fluorines on the central B-ring. The tricyclic fusion oxazolidinone antibacterial agent containing a thiomorpholine group and C head containing a 1,2,3-triazole side chain not only shows excellent antibacterial activity and improved safety but also shows enhanced druggability characteristics, such as high water solubility, good PK and oral bioavailability, excellent metabolic stability, low cytotoxicity, and CYP450 enzyme inhibition, which makes oxazolidinone antibacterials with a linear tricyclic B-C-D ring backbone promising candidates for future research directions [74].

The oxyazolidinone class of antibacterials is relatively new, and linezolid (**22**) and tedizolid (**23**) have been approved by the US FDA. Contezolid (**24**) was approved by the China Food and Drug Administration (CFDA) only in 2021. Contezolid was granted the US FDA’s QIDP classification and granted fast-track status in late 2018. In addition, radezolid (**25**) is a promising oxazolidinone antibiotic with excellent in vivo and in vitro antibacterial activities. Radezolid can effectively inhibit various GPBs and is often used in the treatment of abscesses, bacterial skin diseases, and *Streptococcus* infections. Recently, phase II clinical trials of radezolid for the treatment of pneumonia and uncomplicated skin infections have been completed (NCT00640926, NCT00646958). Structurally, radezolid uses imine as a linker to introduce a triazole ring (D) onto the C ring and thus enhance its interaction with the ribosome (U2584 residues, Figure 8), which increases its efficacy compared with that of linezolid [75].

Linezolid (**22**), (S)-N-[[3-(3-fluoro-4-morpholinylphenyl)-2-oxo-5-oxazolidinyl] methyl] acetamide [76], was the first oxazolidinone antibiotic approved by the FDA for the treatment of hospital-acquired pneumonia (HAP), community-acquired pneumonia (CAP), complex skin infections, and SSTIs caused by MRSA [77]. Linezolid inhibits the synthesis of bacterial proteins by preventing the formation of the ribosomal 70S initiation complex [78]. The SAR analysis of linezolid demonstrated that the morpholine, fluorine atoms, and 5-S configuration are essential for antibacterial efficacy. Introducing an electron-withdrawing group into the aromatic ring (B ring) can increase the antibacterial activity. Introducing a morpholino group (C ring) enhances the PK profile and improves water solubility without affecting the antibacterial activity (Figure 7) [79]. The oral bioavailability and antibacterial activity of linezolid are excellent [80]. However, safety (largely reversible neurotoxicity and bone marrow suppression) has always been a major concern and somewhat limits its application, especially in the community setting [81]. Moreover, due to its reversible MAOI, linezolid sometimes causes toxicity when combined with other drugs; for example, the combination of linezolid with 5-hydroxytryptamine reuptake inhibitors occasionally results in serotonin syndrome [82]. Therefore, improving the safety compared with that of linezolid is a major direction of research on subsequent oxazolidinone drugs.

Tedizolid (**23**) is an active metabolite obtained by phosphatase hydrolysis and approved by the FDA for the treatment of ABSSSIs [83] but was also recently determined to exert potential therapeutic effects against bacteremia and meningitis [84]. Similar to linezolid, tedizolid achieves its antibacterial efficacy by binding to the ribosomal 50S subunit to inhibit the synthesis of bacterial proteins. Tedizolid phosphate is a prodrug of tedizolid that can be converted to active tedizolid by serum phosphatase in vivo [85]. Compared with linezolid, tedizolid phosphate has an improved safety profile (significantly lower risk of thrombocytopenia), an excellent PK profile (high bioavailability and half-life of 11 h), and increased tolerability [86]. However, the hematological toxicity of tedizolid phosphate is currently the major problem limiting its application. Regarding its structure, the introduction of a methyl tetrazole structure (D ring) potentially results in a close interaction between the lone pair of tedizolid phosphate and U2584; the efficacy of tedizolid against *Staphylococcus* and *Enterococcus* is thus 2–8 times that of linezolid [87], and the hydroxyl group at C-5 can form two possible hydrogen bond contacts with the phosphate oxygen and ribose 2′-OH of A2503 (Figure 8A). The introduction of phosphate on the hydroxyl group at C-5 significantly improves the water solubility and bioavailability of the drug compared with those of the parent molecule [88].

Contezolid (**24**) was approved by the CFDA in 2021 for the treatment of cSSTIs and exhibits high activity against Gram-positive pathogens as well as a significantly reduced potential for myelosuppression and monoamine oxidase inhibition (MAOI) [89]. SAR studies have shown that the C-ring dihydropyridone and C-5 isoxazole sidechain work together to promote excellent binding to the ribosome while reducing some side effects, exhibiting activity against Gram-positive pathogens comparable to that of linezolid as well as a significantly reduced potential for myelosuppression and MAOI. Docking studies have shown that trifluorobenzyl (B ring) relies on the hydrophobic pocket defined by the nucleotides C2452 and A2451, and the isoxazole at C-5 forms a π–π interaction with the G2061 residue and a van der Waals interaction with the A2503 residue (Figure 8B) [75]. In addition, a phase II clinical trial in China showed that the safety, compliance, and tolerance of contezolid are better than those of linezolid, and the therapeutic effect of contezolid on MDR bacterial infection is similar to that of linezolid, with fewer hematological adverse events (NCT03747497) [90].

### 3.4. Pleuromutilins

Pleuromutilin antibiotics act on the 50S ribosomal subunit, bind to the V domain of peptidyl transferase, and exert antibacterial effects by blocking the synthesis of bacterial proteins. SAR studies of pleuromutilin antibiotics have shown that the 5-6-8 tricyclic parent nucleus [91], the carbonyl at C-3, and the hydroxyl at C-11 are all needed to achieve antibacterial activity [92]. The C-14 side chain is the main modification site of this class of antibiotics, and chemical modifications of this side chain can improve the antibacterial activity and PK properties. The sulfide and alkaline groups on the side chain of C-14 help improve the antibacterial activity, whereas neutral or acidic groups may reduce the activity of pleuromutilin antibiotics. Currently, the following pleuromutilin antibiotics have been approved for the market: tiamulin (**26**, Figure 9A), valnemulin (**27**), retapamulin (**28**), and lefamulin (**29**). In addition, a phase I clinical trial of azamulin (**30**) was terminated due to low bioavailability [93], but its good selectivity for the CYP3A protein suggests its potential for development as an antitumor drug. BC-3205 (**31**) and BC-7013 (**32**) are two pleuromutilin antibiotics that have completed clinical trials without long-term follow-up studies. Experiments have used microdilution to evaluate the activity of BC-3205 against *S. aureus* and *S. pneumoniae*, and the results show that the MIC ranges from 0.06 to 0.12 µg/mL [94]. BC-7013 exerts a good therapeutic effect on skin infections caused by GPB.

Retapamulin (**28**), the first pleuromutilin to be developed for human topical use [95], is a C14-sulfanyl-acetate derivative of pleuromutilin that contains a tropine sulfide side chain [96]. The in vitro results obtained by microdilution, agar dilution, and e-tests demonstrate that retapamulin has strong activity against resistant *S. aureus* and *S. pyogenes*, with MIC values between 0.06 and 0.125 µg/mL [97]. In addition, retapamulin is a broad-spectrum antibiotic, and the tendency for the development of resistance against retapamulin is low. All these outstanding features have encouraged an increasing number of researchers to pay more attention to pleuromutilin antibiotics.

As a semisynthetic pleuromutilin derivative [98], lefamulin (**29**) was the first antibiotic with a new mechanism that was approved by the FDA for the treatment of CABP in the past two decades [99]. Lefamulin binds to the A site and P site of the peptide transferase center (PTC) of the ribosome through hydrogen bonding to block the process of ribosome translation in bacteria and thereby inhibits the synthesis of bacterial proteins without affecting ribosomal translation in eukaryotes (Figure 9B) [100]. In addition, in vitro studies have shown that this unique mechanism reduces the probability of cross-resistance with other antibiotics [101]. In particular, lefamulin exhibits potent inhibition against *S. aureus* [102]. However, due to the role of the AcrAB-TolC efflux pump, lefamulin has no activity against *Acinetobacter baumannii* (*A. baumannii*), *Enterobacter cloacae* (*E. cloacae*), and *Pseudomonas aeruginosa* (*P. aeruginosa*) [103].

### 3.5. Macrolides

Macrolide antibiotics (MAs) are a class of antibiotics with 12–16-carbon lactone rings in the molecular structure that can irreversibly bind to the 50S subunit of the bacterial ribosome and selectively inhibit the synthesis of bacterial protein by blocking the processes of transpeptidation and mRNA displacement [104]. It is worth noting that the binding sites of MAs on bacterial ribosomes are the same as those of clindamycin and chloramphenicol. Therefore, MAs might appear antagonistic when combined with other antibiotics [105].

Telithromycin (**33**) is the first MA of the ketolide family that was approved for marketing [106]. In terms of its structure, the keto carbonyl at C-3 can reduce the drug resistance rate, and the cyclic carbamate that forms at C-11 and C-12 can increase the affinity for bacterial ribosomes (Figure 10) [107]. Telithromycin can exist stably under acidic conditions and has good activity against penicillin-resistant and macrolide-resistant strains. Moreover, the binding capacity of telithromycin to the ribosomes of wild-type strains is 10 times that of erythromycin and 6 times that of clarithromycin. In terms of PK, telithromycin has a bioavailability of approximately 57% and a half-life of 10 h and can be metabolized by CYP3A4 in the liver. Regrettably, telithromycin potentially causes cardiovascular risk and hepatotoxicity, which limits its potential [108].

Fidaxomicin (**34**) is an MA with a new mechanism and selective antibacterial spectrum that is used in the treatment of *Clostridium difficile*-associated diarrhea (CDAD) in children (older than 6 months of age) and adults [109]. Fidaxomicin rapidly impedes *Clostridium difficile* infection by inhibiting bacterial RNA polymerases [110]. The recurrence rate of *Clostridium difficile* infection treated with fidaxomicin is significantly lower than that obtained with vancomycin. A completed phase III clinical trial of fidaxomicin (NCT01691248) tested the prevention of CDAD in patients who received hematopoietic stem cell transplantation (HSCT) [111]. The results demonstrated that fidaxomicin significantly reduces the incidence of CDAD in HSCT recipients.

## 4. Antibiotics That Interfere with Bacterial Cell Wall Synthesis 

### 4.1. Cephalosporins

Similar to other β-lactam drugs, which bind to bacterial penicillin-binding proteins (PBPs), cephalosporins exhibit antibacterial activity by interfering with the synthesis of peptidoglycan, the main structural component of the bacterial cell wall [112]. Moreover, cephalosporins have low toxicity and do not easily cause allergic reactions [113]. Cephalosporins can be divided into five generations according to their time to market and antibacterial effect. The effect of first-generation cephalosporins on GPBs was stronger than that of second- and third-generation cephalosporins. The second-generation cephalosporins expanded the antibacterial spectrum, and their effectiveness against GNBs was greater than that of the first-generation cephalosporins. Third-generation cephalosporins exert significantly stronger effects against GNBs than first- and second-generation cephalosporins and exhibit strong tissue penetration and high enzyme resistance. Fourth-generation cephalosporins are mainly used for severe infections caused by GNBs with resistance to third-generation cephalosporins. The effect of fifth-generation cephalosporins on GPBs, particularly drug-resistant bacteria, is stronger than that of the first four generations. The SARs of cephalosporins can be summarized as follows (Figure 11): replacement of R_2_ with -CH_3_, -Cl, or a nitrogen-containing heterocyclic ring can improve the absorption, distribution, and metabolism of the drugs in vivo. Replacing S with -O- or -CH_2_- increases the activity and extends the duration of action through the principle of bioelectronics, and the introduction of a benzene ring or a nitrogen-containing heterocyclic ring on the amide group at C-7 expands the antibacterial spectrum. In addition, the introduction of a larger substituent on the same side as the main ring at C-7 improves the stability of β-lactamase and enhances the penetration of the outer membrane of GNB. The quaternary amine group at C-3 forms an internal salt with the carboxyl group in the molecule. C-7 in the main nucleus is connected to a 2-aminothiazole-α-methoxyiminoacetyl side chain. Simultaneous modification of these two positions results in a more potent antibacterial activity compared with that of the unmodified compound (Figure 12A, Table 3).

As a precursor drug of N-phosphine water-soluble cephalosporin, ceftaroline fosamil acetate (**35**) can be transformed into ceftaroline by enzymes in the human body to exert pharmacological effects [114]. In the structure of ceftaroline fosamil acetate, the 1,3-thiazole ring on C-3 is an important group for anti-MRSA activity. The presence of the 1,2,4-thiadiazole ring can significantly improve the affinity of ceftaroline fosamil acetate to transpeptidase and further hinder the synthesis of the bacterial cell wall. Moreover, the presence of a 2-thioimidazole spacer in the side chain can improve the stability and resistance of ceftaroline fosamil acetate.

The C-7 phosphoryl group can improve the solubility of ceftaroline fosamil acetate. In addition, the experimental results obtained by Yuji Lizawa and others have shown that ceftaroline fosamil acetate exerts a stronger effect on MRSA than vancomycin and linezolid (MIC = 2 μg/mL) [115] and is less toxic to the kidneys. In 2016, the product was approved by the FDA and the EMA for the treatment of 2-month-old children and elderly patients with cSSTIs as well as CAP induced by MRSA [116]. In 2019, the EMA authorized the administration of the drug to newborns and infants.

Ceftobiprole medocaril (**36**) is the water-soluble prodrug of ceftobiprole [117] and was approved for the treatment of cSSTIs in 2008. The side chain of methyl pyrrolidone at C-3 in the structure of ceftobiprole medocaril confers high antibacterial activity against GPB, particularly MRSA. A key differentiating factor between ceftaroline and ceftobiprole is binding to the unique PBP in MRSA (PBP 2a) in addition to binding to other PBPs in Gram-positive and Gram-negative bacteria. Three phase III studies of patients with ABSSSIs, CAP, and HAP have been completed, and all treatment groups displayed similar microbiological success and safety profiles (NCT03137173, NCT00326287, and NCT00229008) [118]. At present, Basilea Pharmaceutica is conducting a phase III clinical study to evaluate the efficacy of ceftobiprole medocaril against complex *S. aureus* bacteremia (NCT03138733).

Ceftolozane/tazobactam (**37**, **38**) consists of ceftolozane, a novel *Pseudomonas* cephalosporin, and tazobactam, an established β-lactamase inhibitor [119]. Ceftolozane is a new cephalosporin antibiotic that was obtained by modifying the side chain located at the third position of the parent nucleus of cephem to enhance its antibacterial activity against *P. aeruginosa* [120]. Although tazobactam itself has no antibiotic activity, it can enhance the activity of β-lactam antibiotics to overcome bacterial resistance [121]. As an alternative to ceftazidime/avibactam, ceftolozane/tazobactam provides a promising treatment option for MDR *P. aeruginosa* infections [122].

Similar to ceftolozane/tazobactam, ceftazidime/avibactam (**39**, **40**) is also a combination agent for the treatment of resistant GNB infections [123]. Ceftazidime, which was approved in 1985, was originally a third-generation cephalosporin, and avibactam is a non-β-lactam β-lactamase inhibitor [124]. The activity of this agent is limited against β-lactamase-producing isolates, Gram-negative anaerobes, and GPB. *Acinetobacter* spp. are not considered for ceftazidime/avibactam, mainly due to the inability of avibactam to penetrate the adventitia [125]. Currently, ceftazidime/avibactam is an important drug for the treatment of *P. aeruginosa* infection, but data show that 50.9% of extensively drug-resistant *P. aeruginosa* strains are resistant to ceftazidime/avibactam. Therefore, an exploration of effective antibacterial drug combination therapy strategies is urgently needed. In one study, imipenem combined with ceftazidime/avibactam or avibactam was found to exhibit synergistic effects against metalloenzyme-producing pan-resistant *P. aeruginosa* [126].

Cefiderocol (**41**) is a novel parenteral siderophore cephalosporin that was discovered and developed by Shionogi Inc., Japan [127]. This drug is unique due to its siderophore-like property [128]. The side chain at C-3 of cefiderocol promotes the formation of a chelating complex with ferric iron to promote iron carrier-like transport through the outer membrane of GNB using an iron transport system. Cefiderocol is delivered to the periplasmic space through the outer membrane and released in the periplasmic space, where it binds to PBPs, mainly PBP3, and this binding inhibits peptidoglycan synthesis and leads to cell death (Figure 13). In addition, the entry of iron carriers into the periplasmic space of bacteria enhances the protection of bacterial β-lactamase stability [129].

Cefiderocol contains a catechol moiety, which is necessary for the chelation of iron by cefiderocol with iron and important for the transport of iron into *P. aeruginosa* [130]. Moreover, the carboxylic acid of the C-7 side chain increases the permeability of cefiderocol to the outer membrane, and the chlorocatechol group of the C-3 side chain is iron-chelated [131]. The ability to chelate ferric ions results in an iron-depleted environment, which promotes the absorption of cefiderocol.

### 4.2. Carbapenems

Compared with other β-lactams, carbapenems have the most extensive activity and stability, particularly against MDR Gram-negative pathogens [132]. The structure of carbapenems is similar to that of penicillins and cephalosporins, and these compounds form through condensation of the β-lactam ring and an unsaturated five-membered ring [133]. The differences in their structure include the replacement of the sulfur atom on the thiazolyl ring with a carbon, and the 6-hydroxylethyl side chain is in the trans conformation (Figure 12B). This special configuration confers multiple advantages to β-lactamase, including strong, broad-spectrum antibacterial activity, high stability, and low toxicity. Furthermore, the stereochemistry of the hydroxyethyl side chain is essential for activity [113]. Carbapenems can inhibit the activity of cell wall mucin synthase and thus hinder the formation of cell wall mucin, which results in an incomplete cell wall and further changes the osmotic pressure of the cytoplasm or dissolves the cells to kill bacteria [134]. The carbapenems approved in this century are ertapenem (2002), biapenem (2002), doripenem (2005), tebipenem pivoxil (2009), meropenem/vaborbactam (2017), and imipenem/cilastatin/relebactam (2019).

Ertapenem (**42**) is a carbapenem with a wide antibacterial spectrum and strong antibacterial effect [135]. This drug has several advantageous characteristics, including stability against renal dehydropeptidase, low generation of drug resistance, good clinical treatment effect, good tolerance, few adverse drug reactions (ADRs), and a long half-life [136]. Ertapenem is commonly used to treat complex abdominal infections, complex skin and skin structure infections, CAP, cUTIs, and acute pelvic infections in the clinic [137]. A phase II clinical trial of cefazolin combined with ertapenem in the treatment of methicillin-susceptible *S. aureus* bacteremia is currently ongoing (NCT04886284).

The antibacterial activity of biapenem (**43**) is similar to that of other carbapenems, and its stability against renal dehydropeptidase is better than that of imipenem, meropenem, and panipenem [138]; this agent is used for the treatment of severe infections and extensive drug-resistant infections. Biapenem has almost no nephrotoxicity or nervous system toxicity.

Doripenem (**44**) is a 1-β-methyl carbapenem with high chemical stability [139], and its side chain makes it resistant to hydrolysis by dehydropeptidase-1 [140]. Doripenem shows concentration-independent characteristics against GPB and has better activity against penicillin-susceptible *Streptococci viridans* (MIC_90_ = 0.25 µg/mL) than imipenem (MIC_90_ ≤ 0.5 µg/mL) [141]. In addition, doripenem has a good safety profile despite a certain degree of gastrointestinal discomfort and allergic reactions.

Tebipenem pivoxil (**45**) is an oral carbapenem antibiotic [142]. The key structural feature of tebipenem pivoxil is that the side chain at C-3 is a thiazolyl-substituted azacyclobutane group. Moreover, oral absorption is improved through the formation of a pivoxil prodrug at the C-2 carboxylic acid [143]. The oral absorption of this product is better than that of most β-lactam antibiotic products on the market. Tebipenem pivoxil has a wide antibacterial spectrum. For most clinically isolated strains, tebipenem pivoxil shows stronger antibacterial activity than the penicillin series and cephalosporin series, and when injected, tebipenem pivoxil exerts the same or better antibacterial effect in comparison to the other carbapenem antibiotics [144].

Meropenem/vaborbactam (**46**, **47**) is an antibacterial composite drug composed of meropenem and vaborbactam and is commercially named Vabomere [145]. Meropenem inhibits the synthesis of the cell wall to produce bactericidal effects and can penetrate the cell walls of most GPB and GNB to bind to PBP targets [146]. The other component, vaborbactam, is a β-lactamase inhibitor with no antibacterial activity that can protect meropenem from degradation by some serine-lactamases, such as *Klebsiella pneumoniae* carbapenemase [147]. Vabomere is mainly used in the treatment of cUTIs [148].

Imipenem/cilastatin/relebactam (**48**, **49**, **50**) was approved for the treatment of cUTIs by the FDA in July 2019. Imipenem interferes with cell wall synthesis by binding to the PBPs of *Enterobacteriaceae* and *P. aeruginosa*, and cilastatin can inhibit the metabolism of imipenem in the kidneys and has no antibacterial activity. Relebactam has no antibacterial ability and acts as an enzyme inhibitor to protect imipenem from degradation by some types of serine β-lactamases. Among these compounds, relebactam was approved for the first time as a new molecular entity [149]. A completed phase II randomized controlled trial comparing the safety, tolerability, and efficacy of the combination of 125 mg or 250 mg of relebactam with 500 mg of imipenem with those of imipenem alone showed that the imipenem/relebactam treatment regimen was not inferior to treatment with imipenem alone (NCT01506271) [150].

### 4.3. Cyclic Lipopeptides

Cyclic lipopeptides are synthesized mainly by the microbial nonribosomal synthase pathway and consist of a nonpolar fatty acid chain and a polar amino acid peptide chain, and the amino acid peptide chain forms a ring by itself or in combination with some fatty acid chains (Figure 12C). Cyclic lipopeptides have attracted extensive attention due to their broad-spectrum antibacterial activity, biosurfactant activity, low toxicity, biodegradation ability, and other biological functions.

Daptomycin (**51**) is part of the second generation of glycopeptide antibiotics after vancomycin and is a cyclic ester peptide extracted from *Streptomyces* fermentation broth [151]. Daptomycin has a complex structure with 13 amino acids: 10 are from cyclic lipopeptides, and the remaining 3 are linked to decanoyl groups as side chains [152]. Daptomycin has a hydrophilic nucleus and lipophilic tail, and the lipophilic tail is calcium-dependent, which causes daptomycin to act on the plasma membrane; thus, daptomycin has broad-spectrum antibacterial activity against GPB, and acylation of the primary amine of the ornithine residue results in the better tolerance of daptomycin compared with similar drugs [153]. In addition to acting on the plasma membrane, daptomycin can conduct ions to cause potassium ion outflow and membrane potential dissipation. As a result, macromolecules can be wrapped in the cytoplasm, which makes it difficult for them to flow out, causing cell death [152]. The inhibition of lipid phosphate synthesis is considered another mechanism of daptomycin [154]. This process is used to treat cSSTIs, such as abscesses, surgical incision infections, and skin ulcers, caused by some daptomycin-sensitive Gram-positive strains [155]. Daptomycin is safer and more effective than vancomycin [156]. The incidence of side effects such as nausea and headache is low, but there is a risk of increased phosphokinase after withdrawal [157].

## 5. Antifungal Drugs That Inhibit Cell Membrane Synthesis

### 5.1. Echinocandins

As cyclic lipopeptide antifungal agents [158], echinocandins have advantages including safety, a wide antibacterial spectrum, a long half-life, and a high binding rate to serum proteins. In addition, as glucan synthase inhibitors, echinocandins change the permeability of the fungal cell wall, prevent the successful separation of daughter cells and mother cells, inhibit the formation of mycelia, and lead to cell death after dissolution [159]. Unlike fungal cells, mammalian cells do not have cell walls; thus, echinocandins can act selectively on fungal cells with few side effects on the human body. However, due to their relatively large molecular weight and low oral bioavailability, the options for the administration of these preparations are greatly limited, and these compounds can be provided only through injection. To date, the launched echinocandins include caspofungin, micafungin, and anidulafungin. Structurally, caspofungin has a fatty acid chain, micafungin possesses a 3,5-diphenyl-substituted isoxazole ring side chain, and anidulafungin has a pentyloxyterphenyl side chain. Other structural differences include the amino group in caspofungin and the sulfate group in micafungin (Figure 14A, Table 4) [160]. The different side chains of the three drugs determine their differences in solubility, toxicity, and antibacterial activity.

Caspofungin (**52**) is the first echinomycin drug with a special core structure of cyclic hexapeptide [161]. Caspofungin is a semisynthetic derivative of *S. pneumoniae B* that exhibits substantial improvements in solubility and stability; in particular, the ability to protect against *Aspergillus* species is enhanced by the interconversion of the hemiaminal into an N-ethyl aminal and a reduction in the hydroxyglutamin terminus to the amine [162]. Caspofungin has the largest amount of data regarding patient use and can be used for the empirical treatment of children and adults with *Candida* and *Aspergillus* infections [163]. Caspofungin is recommended as an A-l level drug for empirical treatment by major guidelines.

Micafungin (**53**) is the second echinocandin drug after caspofungin [164]. This drug has good solubility because of a sulfate group at the dihydroxyhomotyrosine, and the fatty N-acyl side chain improves its potency. As a water-soluble semisynthetic lipopeptide, micafungin can selectively inhibit the synthesis of (1-3)-β-d-glucan, and this inhibition directly affects the integrity of the fungal cell wall and leads to cell rupture [165]. Micafungin is mainly used for infections caused by *Candida* [166], but its use is limited due to the risk of hepatotoxicity.

Anidulafungin (**54**) is obtained by diacylation of the echinocandin parent nucleus and introduction of an amino group in the acyl side chain; terphenyl acid is then used to introduce an alkoxy triphenyl group to the amino group, which can help anidulafungin embed in the cell wall and thus improve its effect. However, the side chain has obvious lipophilicity that decreases the water solubility of the drug, which is a disadvantage of this type of drug [167].

### 5.2. Triterpenoids

Ibrexafungerp (**55**, Figure 14B, Table 4) is the first triterpenoid and a fourth-generation, enfumafungin-derived antifungal drug with a broad spectrum that has been approved for the treatment of vulvovaginal candidiasis [168] and displays high antibacterial activity against *Candida parapsilosis* (MIC_90_ = 0.5 μg/mL), *Candida tropicalis* (MIC_90_ = 1.0 μg/mL), *Candida albicans* (MIC_90_ = 1.0 μg/mL), and *Candida krusei* (MIC_90_ = 2.0 μg/mL). This drug was obtained by optimizing the structure of enfumafungin [169] to compensate for the instability of echinocandin in acidic environments, which results in good stability in vivo and in vitro [170].

### 5.3. Triazoles

Triazole antifungal drugs have broad-spectrum activity, good tolerance, and low toxicity and are commonly used for the treatment of invasive fungal infections [171]. These drugs inhibit the biological activity of sterol 14α-demethylase and hinder the synthesis of ergosterol by acting with the CYP450 enzyme system encoded by the fungal CYP51 gene. The accumulated methylated sterols have a certain degree of toxicity because they interfere with the synthesis of cell membrane lipids and thereby change the permeability of cells and cause cell death. Since the 21st century, three known triazole drugs, including the second-generation triazole antifungal drug voriconazole, which was listed in 2002, posaconazole, which was listed in 2006, and the new triazole antifungal drug isavuconazole, which was listed in 2015, have been developed (Figure 14C, Table 4).

Voriconazole (**56**) is a derivative of the first-generation triazole drug fluconazole [172]. Compared with fluconazole, voriconazole has pyrimidine fluoride instead of the triazole ring and α-methyl added, which expands its antibacterial spectrum. Voriconazole can be used for the treatment of invasive aspergillosis [173]. However, as the number of uses increases, the risk of skin malignancy, a serious adverse reaction to voriconazole, also increases [174]. A number of clinical trials have been conducted in many regions to study the clinical outcomes of voriconazole, and these focused mainly on its antitumor, antipneumonia, and antibacterial effects.

Posaconazole (**57**) is a derivative of itraconazole that was developed in 1992 and can be used to prevent and treat invasive aspergillosis [175]. Posaconazole has a wider antibacterial spectrum than voriconazole. The activity of posaconazole (MIC_90_ ≤ 1 µg/mL) against *Candida* in vitro is 2–4 times higher than that of itraconazole [176]. The inhibition rate of posaconazole against *Cryptococcus neoformans* is higher than 99% [177], and more than 95% of filamentous fungi are inhibited [178]. In addition to its broad-spectrum characteristics, the inhibitory effect of posaconazole on the demethylation of ergosterol C-14 is stronger, particularly for *Aspergillus*. Posaconazole has a long side chain added to the parent ring of triazole drugs. This modification increases the affinity of posaconazole for CYP51 and enhances its antibacterial activity. In terms of PK properties, posaconazole can reach an effective concentration in many tissues, and the affinity to alveolar epithelium is good, which makes posaconazole more advantageous for pulmonary fungal infections [179].

Isavuconazole (**58**) is the active metabolite of the water-soluble prodrug isavuconazole sulfate and is obtained by plasma esterase conversion after isavuconazole sulfate enters the human body [180]. Isavuconazole is currently the only triazole antifungal agent for which a large-scale controlled study with *amphotericin B* has been completed worldwide. This drug inhibits the 14-α demethylation of lanosterol, and its broad-spectrum activity is exerted by the side arm of isavuconazole such that the triazole ring can bind to the pocket of the ferrous heme group [181]. This structure makes isavuconazole more effective than other triazole drugs. It has good tolerance, and phase I and phase II clinical trials have shown that isavuconazole has no serious side effects (NCT01657890, NCT00413439). However, the common symptoms include slight abdominal pain and vomiting due to slight nausea.

## 6. Antituberculosis Drugs

Tuberculosis is a serious infectious disease mainly caused by *Mycobacterium tuberculosis*, which is mainly transmitted by breathing. In the treatment of tuberculosis, multidrug-resistant *Mycobacterium tuberculosis* and extensively drug-resistant *Mycobacterium tuberculosis* constitute a huge obstacle for the drug prevention and treatment of tuberculosis. The current clinical treatment of pulmonary tuberculosis has many shortcomings, such as a long duration of drug treatment, poor compliance, substantial toxicity and side effects, and a lack of medication for children. There is an urgent need for new antituberculosis drugs with high efficacy, low toxicity, low cost, and a short treatment period for multidrug-resistant Tubercle Bacillus (MDR-TB). At present, conventional antituberculosis drugs easily develop drug resistance and have obvious side effects; thus, the research and development of new antituberculosis drugs are ongoing. Three new antituberculosis drugs have been approved by the FDA: bedaquiline (**59**), delamanid (**60**), and pretomanid (**61**). Bedaquiline exerts its bactericidal effect by inhibiting the adenosine triphosphate (ATP) synthase of *Mycobacterium tuberculosis*, and delamanid and pretomanid exert their antibacterial effect by inhibiting the synthesis of the cell wall of *Mycobacterium tuberculosis* (Figure 15, Table 5).

Bedaquiline (**59**) is a diarylquinoline drug that has been approved for the treatment of MDR-TB for more than 50 years [182]. Bedaquiline contains a quinolinic central heterocyclic nucleus with alcohol, and its anti-TB activity is attributed to the diarylquinoline ring, the side chain with the N,N-dimethyl amino terminus, the hydroxyl group, and the naphthalene moiety [183]. As an additional antimycobacterial agent, bedaquiline is a great addition [184]. This compound can specifically inhibit ATP synthase by binding to the C subunit and thus interferes with energy production and homeostasis in the cell. Bedaquiline is primarily oxidatively degraded by CYP3A4 in the liver to N-desmethyl metabolite. The anti-*Mycobacterium tuberculosis* (MTB) activity of the N-desmethyl metabolite is 20% to 60% of that of bedaquiline and has no obvious effect, but the increase in its plasma concentration may lead to prolongation of the QT interval [185]. However, bedaquiline has the advantages of high selectivity, good activity, long-lasting efficacy, and almost no toxicity or side effects and exhibits good activity against multidrug-resistant and extensively drug-resistant tuberculosis.

Delamanid (**60**), a nitro-dihydro-imidazooxazole derivative, was designed and synthesized by Japan’s Otsuka Pharmaceutical Co., Ltd., based on the study of the SARs of compounds of the antituberculosis candidate drug PA-824 series [186]. The drug interferes with the metabolism of the MTB cell wall by inhibiting the synthesis of mycolic acid (an important part of the waxy outer shell of MTB, which can help the bacteria resist penicillin and most other antibiotics) and then inhibits activity against drug-resistant strains [187]. Notably, the most commonly reported adverse reactions observed with delamanid are nausea, vomiting, and dizziness. The use of delamanid with antiretrovirals that prolong the QT interval requires monitoring [188].

Pretomanid (**61**) is a new type of compound belonging to the nitroimidazole class of drugs. This class of compounds targets decaprenylphosphoryl-β-D-ribose-2’-epimerase-1 to inhibit bacterial cell wall synthesis and exert antituberculosis activity [189]. Pretomanid, bedaquiline, and linezolid together form the BPaL (B: bedaquiline; Pa: pretomanid; L: linezolid) regimen for the treatment of refractory drug-resistant pulmonary tuberculosis, which involves a short course of treatment and has a high cure rate, including against extensively drug-resistant tuberculosis and multidrug-resistant tuberculosis for which there are no effective chemotherapy regimens [190]. Compared with traditional regimens, the BPaL regimen can treat MDR-TB patients with cost savings in an environment with a high burden of drug-resistant TB, which enables the rapid clinical application of the BPaL regimen to address the significant programmatic and clinical challenges faced by countries with high drug-resistant TB burdens in managing patients with MDR-TB [191].

## 7. Conclusions and Perspectives

As the most numerous low-grade organisms, bacteria have a strong ability to adapt to various environments and can avoid being harmed by antibiotics through self-evolution and mutation. Thus, the development of new antibiotics has always been a constant task performed by medicinal chemists. It is worth noting that antibiotics with new antibacterial mechanisms and novel structures are constantly being discovered, which has addressed the problem of antibiotic resistance in the last two decades.

As mentioned above, quinolones account for the largest proportion of new antibiotics, partly because the SARs of quinolones are relatively well established. In recent years, the identification of quinolones with enhanced activity against problematic Gram-positive organisms, such as delafloxacin (**10**), has been the principal focus. As a chemical species that is already widely used clinically, aminoglycosides play an important role. Plazomicin (**21**), which was developed by modifying sisomicin (**21’**), has good curative effects in the treatment of cUTIs, but its potential toxicity still limits its clinical utilization. Tetracycline drugs, such as omadacycline (**19**), can simultaneously treat CABP and ABSSSI and can prevent drug resistance due to efflux pump and ribosome protection mechanisms, particularly in the treatment of infections caused by *Legionella pneumophila*, *Mycoplasma pneumoniae*, and *Chlamydia pneumoniae*. Therefore, omadacycline can be used as an alternative to the empirical combination of β-lactam and macrolide for the treatment of CABP. As a potent oxazolidinone drug, linezolid (**22**) has been used for many years with great commercial success, which has prompted many pharmaceutical companies to dedicate resources to the development of oxazolidinone antibiotics. To date, four main types of chemical modifications of oxazolidinone antibacterial agents have been identified: modification of the oxazolidinone, phenyl, morpholine rings, and C-5 side chains [192]. Tedizolid (**23**), which has a C-5 hydroxymethyl moiety, shows good activity against linezolid-resistant strains due to methylation of the chloramphenicol–florfenicol resistance (Cfr) gene, which makes it difficult for the acetamide of linezolid to bind to the target through steric hindrance and results in drug resistance. The replacement of acetamide with a smaller hydroxyl group overcomes the steric hindrance caused by Cfr-mediated methylation; thus, tedizolid is more effective in treating resistant bacterial infections [193]. The morpholine ring-modified derivative radezolid (**25**) significantly overcomes the most common ribosomal mutation clinically associated with linezolid resistance (G2576U). In addition, the ability of radezolid to replace chloramphenicol or puromycin is dozens of times higher than that of linezolid, and this finding is related to the stronger ability of radezolid to bind to ribosomes and cause translational inaccuracies [194]. Contezolid (**24**) is approved for the treatment of complicated skin infections and soft tissue infections. Although oxazolidinones contribute greatly to the treatment of resistant bacterial infections, their side effects, including serotonin syndrome, lactic acidosis, thrombocytopenia, and peripheral and optic neuropathy, which are related to mitochondrial toxicity, remain a clinical concern. It is expected that good candidates can likely be screened from C-5 inverted amide analogs and oxazolidinone derivatives with substituted piperidine or azetidine C rings, which can decrease the MAOI. In addition to these classical categories, pleuromutilins have offered a new avenue to explore in recent years because pleuromutilin analogs can be easily obtained by modifying C-14 glycolate with thiol ester while maintaining the unmodified tricyclic scaffold. The hybridization of pleuromutilin and other advantageous antibacterial fragments can be a successful strategy to cope with resistance. Pyridine- or thiazole-substituted benzenesulfonamide improves the elimination of *S. aureus*, *Streptococcus equi*, and *Mycoplasma gallisepticum* compared with that achieved with the benzoxy-pleuromutilin and sulfonamide lead compounds. The slower development of resistance observed with pleuromutilins is another advantage and is not effluxed by ATP-binding cassette (ABC) transporters. The most common side effects of pleuromutilins are injection site stimulations, diarrhea, nausea, vomiting, and elevated transaminase. Individuals with arrhythmia or other prolonged QT intervals should also avoid using pleuromutilins.

The splicing of two or more antibacterial privileged scaffolds via combination principles is also an effective route for the development of new antibiotics. The combination of quinolone and oxazolidinone is a good example, partly because their clinical antibacterial spectra are complementary. Specifically, cadazolid is an oxazolidinone-quinolone hybrid that demonstrated potent antibacterial efficiency in a phase III clinical trial (NCT01983683). Most drugs that are currently on the market are eligible for GPB treatment, but the increasing clinical demand for GNB antibiotics is an unmet need [195].

Notably, surveys of search strategies for new antibiotics that rely on the chemical modification of natural products (semisynthesis) are insufficient to combat the threats of resistance that are rapidly evolving. In addition, semisynthesis products have a limited scope for improvement: they usually increase the molecular weight of scaffolds and rarely permit modification of the underlying scaffold. Mitcheltree et al. [196] reported the component-based total synthesis of a rigid oxepanoproline scaffold that, when linked to the aminooctose residue of clindamycin, formed an efficient broad-spectrum antimicrobial agent named iboxamycin. This novel antibacterial agent constitutes a substantive rescaffolding of the lincosamide antibiotics and is effective against multiple multiantibiotic-resistant pathogens. This study highlighted the contribution of chemical synthesis to supplying antibiotics that can overcome the increasing problems with MDR bacteria worldwide.

Rational antibiotic design depends heavily on a thorough understanding of the targets at the molecular level. The cocrystal structure of the target protein and the small-molecule drug ligand is the most critical information in drug design. Cocrystal structures can not only clarify the SARs and reveal the binding mode, bioactive conformation, and new binding pockets or allosteric binding sites but also open up new horizons for drug discovery, such as fragment-based screening, rational design of compound libraries, and de novo identification of new skeletons. Based on this method, many pharmaceutical companies in the United States have jointly developed a new class of antibiotics inspired by the bacterial β-lactamase inhibitor diazabicyclooctane. Such compounds can inhibit bacterial β-lactamase and bacterial PBP2 and exhibit good in vitro antibacterial activity. Among a series of derivatives, the compound ETX0462 has good target selectivity and permeability, which makes it effective against drug-resistant bacteria with efflux pump mutations. This perspective provides new ideas and methods for the structural optimization of drug derivatives [197].

Although the concepts of computer-aided drug design and rational drug design have been thoroughly explored by medicinal chemists and despite the involvement of artificial intelligence and machine learning in the development of new antibiotics, the identification of novel antibiotics with high efficiency and low toxicity remains difficult. Therefore, under the guidance of the released SARs, synthesizing a large number of novel derivatives and conducting large-scale screening of clinically isolated drug-resistant strains after multiple rounds of optimization is an effective method for obtaining candidate antibiotics for clinical study, even though this process is time-consuming and labor-intensive.

Researchers can develop new antibiotics using some new strategies. One of the strategies is modifying the original chemical structure of an antibiotic to circumvent the antibiotic resistance mechanism. For example, in tigecycline (**17**), a derivative of minocycline, a 9-tert-butyl-glycyl side chain has been added to the D ring at the ninth position of the molecule, which results in avoiding the specific tetracycline efflux pump or the role of ribosome protection [198]. The second strategy is the development of compounds that inhibit antibiotic resistance mechanisms, such as the development of new β-lactamase inhibitors [199]. In addition, abundant bacterial genome sequence information can provide clues for the discovery of new antibiotics. This approach is achieved by targeting specific virulence factors involved in this process to influence pathogenicity. In terms of materials, nanoparticles can achieve sterilization by producing bactericidal components. The possible mechanisms of action include the induction of oxidative stress, nonoxidative mechanisms, and the interaction of released metal ions with functional groups of proteins and nucleic acids. It is expected that with the joint efforts of doctors, pharmaceutical companies, countries, and the public around the world, an appropriate strategy to solve antibiotic resistance will surely be found.

## Figures and Tables

**Figure 1 molecules-28-01762-f001:**
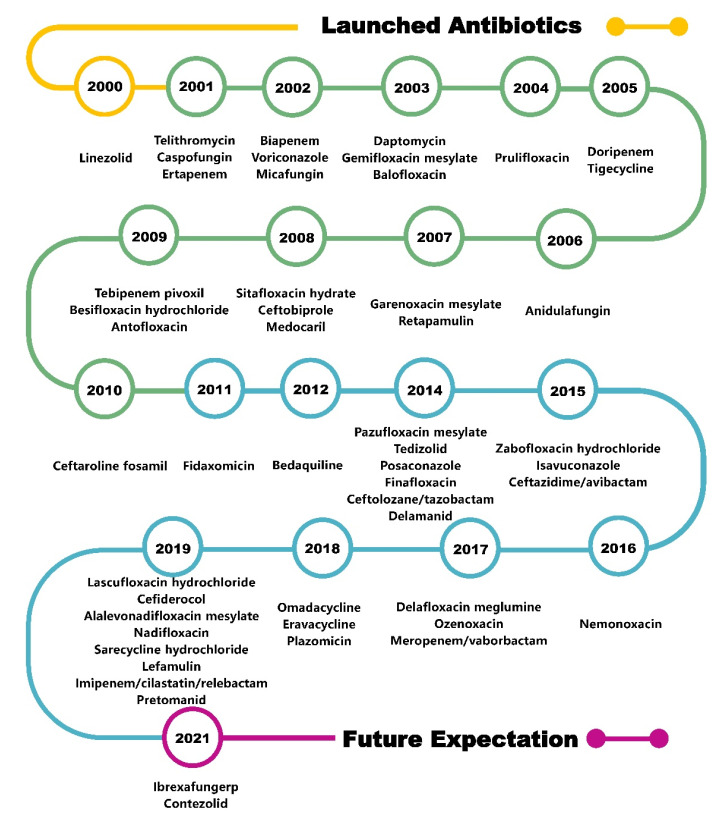
Timeline of antibiotics launched in the last two decades.

**Figure 2 molecules-28-01762-f002:**
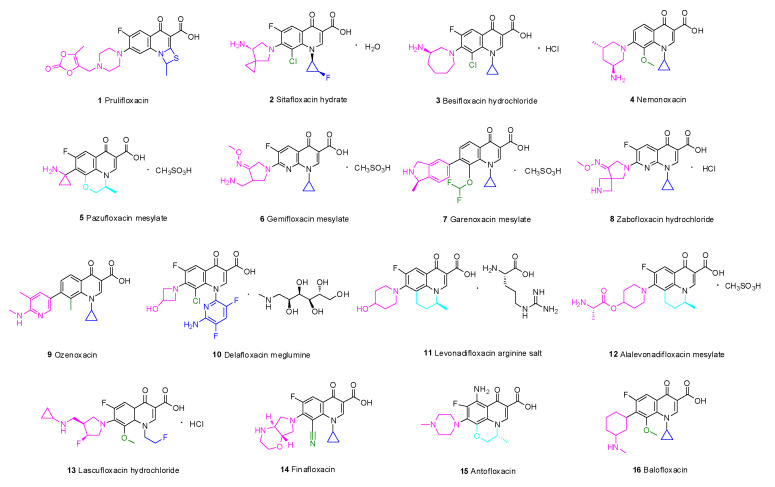
Quinolones launched in the last two decades.

**Figure 3 molecules-28-01762-f003:**
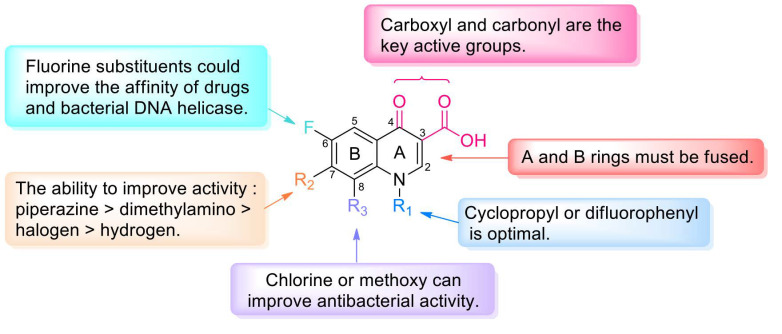
Structure–activity relationships of quinolones.

**Figure 4 molecules-28-01762-f004:**
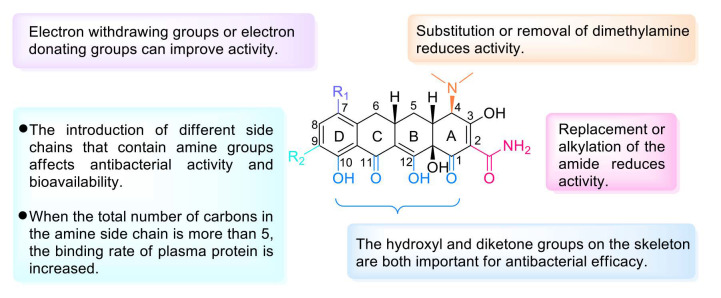
Structure–activity relationships of tetracyclines.

**Figure 5 molecules-28-01762-f005:**
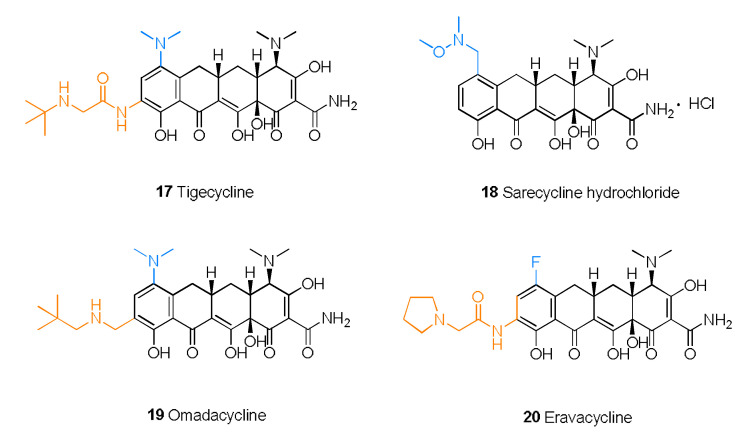
Structures of tetracyclines **17**–**20**.

**Figure 6 molecules-28-01762-f006:**
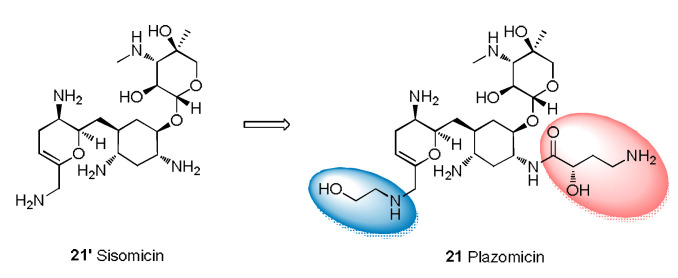
Structural modification from sisomicin to plazomicin.

**Figure 7 molecules-28-01762-f007:**
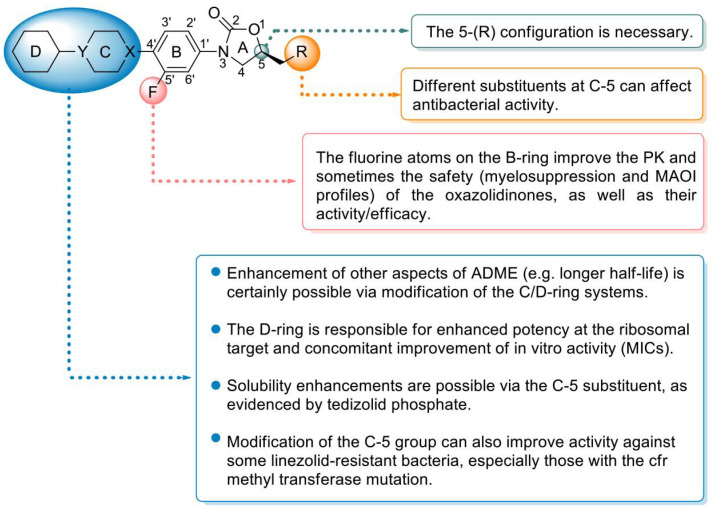
Structure–activity relationships of oxazolidinone antibiotics.

**Figure 8 molecules-28-01762-f008:**
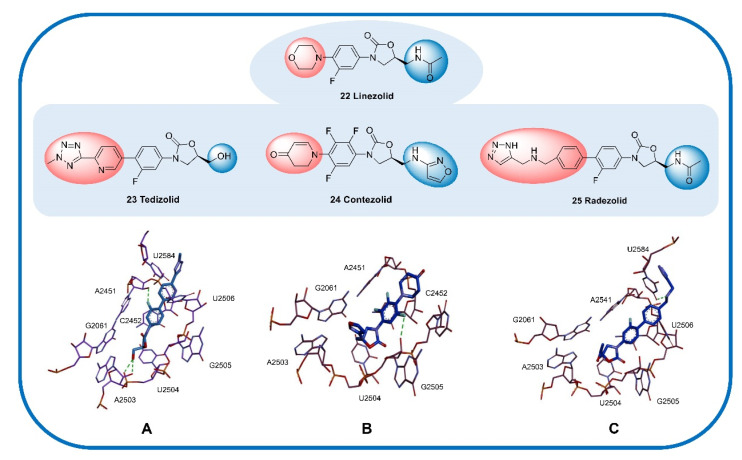
Cocrystal structure of some oxazolidinone antibiotics with residues in the 50S ribosomal subunit. (**A**) Tedizolid (PDB code: 6WRS). (**B**) Contezolid (PDB code: 6WQN). (**C**) Radezolid (PDB code: 6WQQ). The compounds are shown in blue, and the hydrogen bonds are shown as green lines.

**Figure 9 molecules-28-01762-f009:**
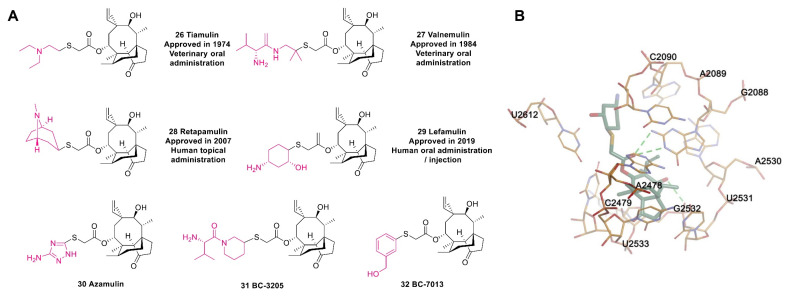
(**A**) Structures of pleuromutilin antibiotics. (**B**) Cocrystal structure of lefamulin binding to the ribosomal 50S subunit of *Staphylococcus aureus* (PDB code: 5HL7).

**Figure 10 molecules-28-01762-f010:**
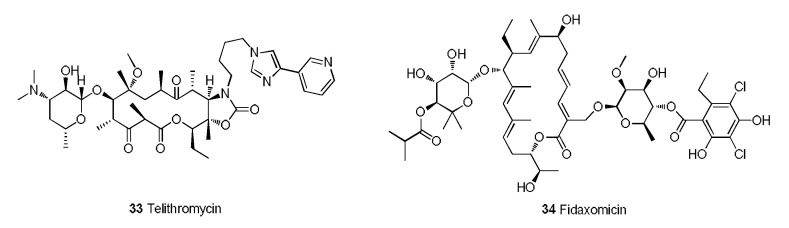
Telithromycin and fidaxomicin.

**Figure 11 molecules-28-01762-f011:**
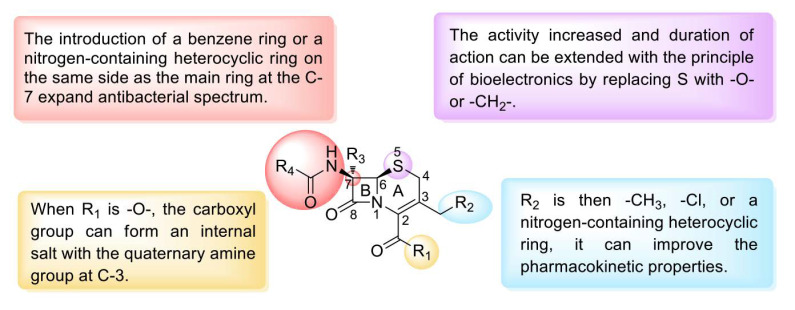
Structure–activity relationships of cephalosporins.

**Figure 12 molecules-28-01762-f012:**
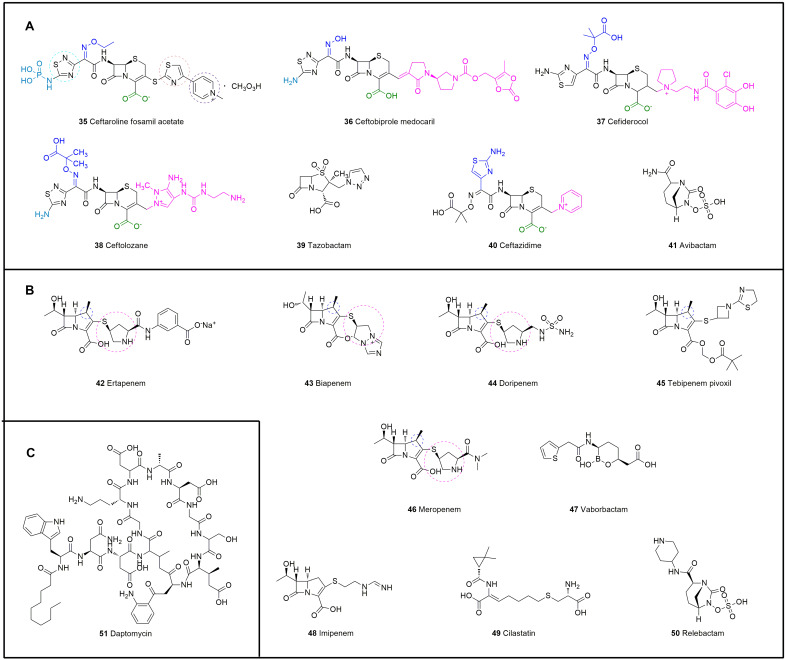
Chemical structures of antibiotics that interfere with the synthesis of bacterial cell walls. (**A**) Cephalosporins, **35**–**41**. (**B**) Carbapenems, **42**–**50**. (**C**) Daptomycin, **51**.

**Figure 13 molecules-28-01762-f013:**
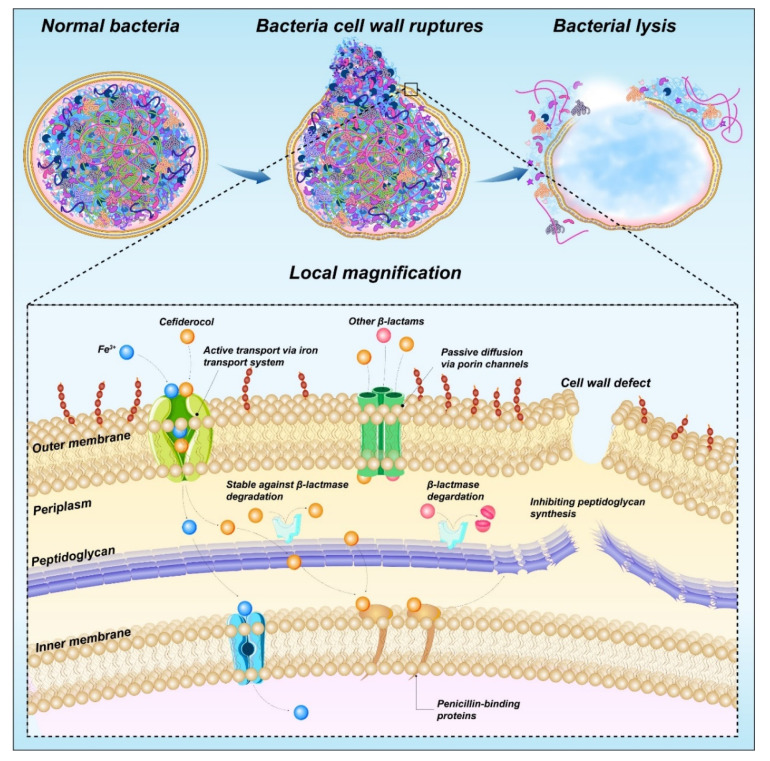
Mechanism of action of cefiderocol.

**Figure 14 molecules-28-01762-f014:**
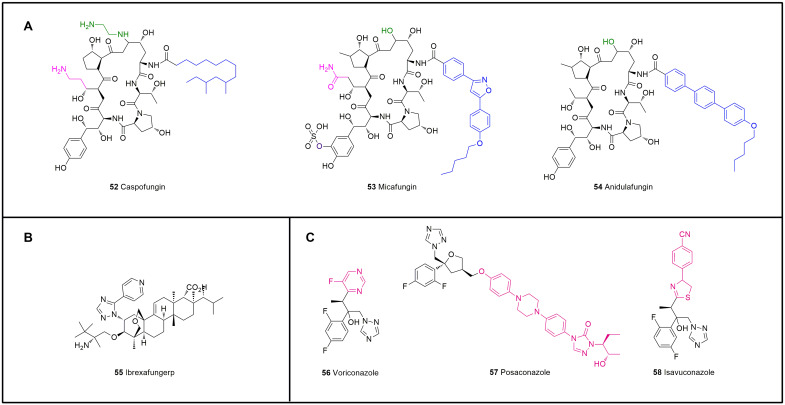
Structures of antifungal drugs that inhibit cell membrane synthesis. (**A**) Echinocandins, **52**–**54**. (**B**) Ibrexafungerp, **55**. (**C**) Triazoles, **56**–**58**.

**Figure 15 molecules-28-01762-f015:**
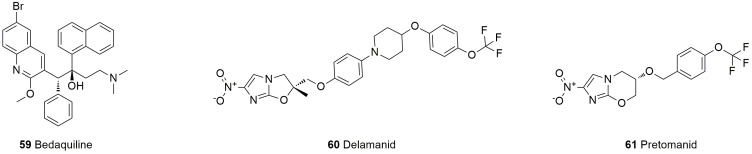
Structures of antituberculosis drugs.

**Table 1 molecules-28-01762-t001:** Summary of launched DNA topoisomerase inhibitors.

Generic Name	Brand Name	Company	Approval	CAS Number	Clinical Indications	Molecular Mechanism
Prulifloxacin (**1**)	Pruvel	Nippon Shinyaku	2004	123447-62-1	Bronchitis Respiratory tract infections Gastroenteritis	DNA gyrase inhibitor
Sitafloxacin hydrate (**2**)	Gracevit	Daiichi Pharmaceutical	2008	163253-36-9	Gram-negative bacterial infection Pneumonia	DNA gyrase inhibitor, DNA topoisomerase IV inhibitor
Besifloxacin hydrochloride (**3**)	Besivance	Bausch & Lomb	2009	141388-76-3	Ocular bacterial infection	DNA gyrase inhibitor, DNA topoisomerase IV inhibitor
Nemonoxacin (**4**)	Taigexyn	Procter & Gamble	2016	378746-64-6	Diabetic complications Pneumonia Gram-positive bacterial infection	DNA topoisomerase IV inhibitor
Pazufloxacin mesylate (**5**)	Pasil	FUJIFILM Toyama Chemical	2014	127046-45-1	Conjunctivitis, otitis media Pneumonia	DNA gyrase inhibitor, DNA topoisomerase IV inhibitor
Gemifloxacin mesylate (**6**)	Factive	LG Chem	2003	210353-56-3	Acute rhinosinusitis Gram-negative bacterial infection Pneumonia	DNA gyrase inhibitor, DNA topoisomerase IV inhibitor
Garenoxacin mesylate (**7**)	Geninax	Astellas & Taisho Toyama	2007	223652-81-1	Respiratory tract infections Pneumonia	DNA gyrase inhibitor, DNA topoisomerase IV inhibitor
Zabofloxacin hydrochloride (**8**)	Zabolante	Dong Wha Pharmaceutical	2015	219680-11-2	Bronchitis Gram-positive bacterial infection Pneumonia	DNA gyrase inhibitor, DNA topoisomerase IV inhibitor
Ozenoxacin (**9**)	Ozanex	Medimetriks Pharmaceutical	2017	245765-41-7	Acne Impetigo	DNA gyrase inhibitor, DNA topoisomerase IV inhibitor
Delafloxacin meglumine (**10**)	Baxdela	Rib-X Pharmaceutical	2017	352458-37-8	Gonorrhea Acute bacterial skin and skin structure infections	DNA gyrase inhibitor, DNA topoisomerase IV inhibitor
Levonadifloxacin arginine salt (**11**)	Emrok	Wockhardt	2019	371246-52-5	Acute bacterial skin and skin structure infections	DNA gyrase inhibitor, DNA topoisomerase IV inhibitor
Alalevonadifloxacin mesylate (**12**)	Emrok O	Wockhardt	2019	306303-00-4	Acute bacterial skin and skin structure infections	DNA gyrase inhibitor, DNA topoisomerase IV inhibitor
Lascufloxacin hydrochloride (**13**)	Lasvic	Kyorin	2019	848416-07-9	Pneumonia	DNA gyrase Inhibitor, DNA topoisomerase IV inhibitor
Finafloxacin (**14**)	Xtoro	MerLion Pharmaceutical	2014	209342-40-5	Acute otitis externa	DNA gyrase inhibitor, DNA topoisomerase IV inhibitor
Antofloxacin (**15**)	Youpeng	Anhui Global Pharmaceutical	2009	119354-43-7	BronchitisAcute pyelonephritis	DNA gyrase inhibitor
Balofloxacin (**16**)	Qroxin	Roche	2003	127294-70-6	Urethritis Urinary tract infection	DNA gyrase inhibitor

**Table 2 molecules-28-01762-t002:** Summary of protein synthesis inhibitors that act on ribosomal subunits.

Generic Name	Brand Name	Company	Approval	CAS Number	Clinical Indications	Molecular Mechanism
Tigecycline (**17**)	Tygacil	Pfizer	2005	220620-09-7	Acute bacterial skin and skin structure infections Gram-negative bacterial infection	Ribosomal protein inhibitor
Sarecycline hydrochloride (**18**)	Seysara	Paratek Pharmaceutical	2019	1035979-44-2	Anthrax, cystitis, rosacea	Ribosomal protein inhibitor
Omadacycline (**19**)	Nuzyra	Paratek Pharmaceutical	2018	389139-89-3	Acute bacterial skin and skin structure infections	Ribosomal protein inhibitor
Eravacycline (**20**)	Xerava	Tetraphase Pharmaceutical	2018	1207283-85-9	Complicated urinary tract infection MDR bacterial infection	Ribosomal protein inhibitor
Plazomicin (**21**)	Zemdri	Achaogen	2018	1380078-95-4	Complicated urinary tract infection Gram-negative bacterial infection Pyelonephritis	Ribosomal protein inhibitor
Linezolid (**22**)	Ozanex	Pfizer	2000	245765-41-7	Skin and soft tissue infections Bacterial pneumonia, tuberculosis	Monoamine oxidase inhibitor Ribosomal protein inhibitor
Tedizolid (**23**)	Sivextro	Dong-A Pharmaceutical	2014	856867-55-5	Acute bacterial skin and skin structure infection	Breast cancer-resistant protein inhibitor Monoamine oxidase inhibitor
Contezolid (**24**)	MRX-I	MicuRx	2021	1112968-42-9	Complex skin and soft tissue infections	Monoamine oxidase inhibitor
Retapamulin (**28**)	Altabax	GlaxoSmithKline	2007	224452-66-8	Phagocytosis caused by *Staphylococcus aureus* or *Streptococcus pyogenes*	Ribosomal protein inhibitor
Lefamulin (**29**)	Xenleta	Nabriva	2019	1061337-51-6	Acute bacterial skin and skin structure infections Pneumonia	Ribosomal protein inhibitor
Telithromycin (**33**)	Ketek	Sanofi	2001	191114-48-4	Pneumonia	Ribosomal protein inhibitor
Fidaxomicin (**34**)	Dafclir	Optimer	2014	56645-60-4	*Clostridioides difficile*-associated diarrhea infection MDR bacterial infection	DNA-directed RNA polymerase inhibitor

**Table 3 molecules-28-01762-t003:** Summary of drugs that interfere with bacterial cell wall synthesis.

Generic Name	Brand Name	Company	Approval	CAS Number	Clinical Indications	Molecular Mechanism
Ceftaroline fosamil acetate (**35**)	Teflaro	Takeda	2010	229016-73-3	Osteomyelitis, sepsis, bacteremia Acute bacterial skin and skin structure infections, pneumonia	Penicillin-binding protein inhibitor
Ceftobiprole medocaril (**36**)	Mabelio	Johnson & Johnson & Roche	2008	376653-43-9	Bacteremia, neutropenia, pneumonia Acute bacterial skin and skin structure infections	Penicillin-binding protein inhibitor
Ceftolozane/tazobactam (**37**, **38**)	Zerbaxa	Astellas	2014	1613740-46-7	Complicated urinary tract infection Complicated intra-abdominal infection	β-lactamase inhibitor
Ceftazidime/avibactam (**39**, **40**)	Avycaz	Sanofi	2015	1393723-27-7	Complicated urinary tract infection Complicated intra-abdominal infection	Non-β-lactam β-lactamase inhibitor combination for serious Gram-negative infection
Cefiderocol (**41**)	Fetcroja	Shionogi	2019	1225208-94-5	Complicated urinary tract infection Pyelonephritis	Penicillin-binding protein inhibitor
Ertapenem (**42**)	Invanz	AstraZeneca	2001	153773-82-1	Endometritis Acute bacterial skin and skin structure infections	Penicillin-binding protein inhibitor
Biapenem (**43**)	Omegacin	Pfizer	2002	120410-24-4	Sepsis, pneumonia, lung abscess	β-lactamase inhibitor
Doripenem (**44**)	Doribax	Shionogi	2005	148016-81-3	Bacterial meningitis Fibrosis, pneumonia	Dehydropeptidase I inhibitor Penicillin-binding protein inhibitor
Tebipenem pivoxil (**45**)	Orapenem	Pfizer	2009	161715-24-8	Gram-negative bacterial infection	Penicillin-binding protein inhibitor
Meropenem/vaborbactam (**46**, **47**)	Vabomere	Novartis	2017	2031124-72-6	Complicated urinary tract infection	β-lactamase inhibitor
Imipenem/cilastatin/relebactam (**48**, **49**, **50**)	Recarbrio	Merck	2019	1174018-99-5	Complicated urinary tract infection Complicated intra-abdominal infection	Non-β-lactam β-lactamase inhibitor combination for serious Gram-negative infection
Daptomycin (**51**)	Cubicin	Lilly	2003	103060-53-3	*Staphylococcus aureus*-caused bloodstream infection accompanied by infective endocarditis	Peptidoglycan synthesis inhibitor

**Table 4 molecules-28-01762-t004:** Antifungal drugs that inhibit cell membrane synthesis.

Generic Name	Brand Name	Company	Approval	CAS Number	Clinical Indications	Molecular Mechanism
Caspofungin (**52**)	Cancidas	Merck	2001	162808-62-0	Invasive aspergillosis	Glucan synthase inhibitor
Micafungin (**53**)	Mycamine	Astellas	2002	235114-32-6	Candidiasis Severe systemic infection	Glucan synthase inhibitor
Anidulafungin (**54**)	Eraxis	Lilly	2006	166663-25-8	Adult candidiasis	Glucan synthase inhibitor
Ibrexafungerp (**55**)	Brexafemme	Merck	2021	1965291-08-0	Vulvovaginal candidiasis	Glucan synthase inhibitor
Voriconazole (**56**)	Vfend	Pfizer	2002	137234-62-9	Invasive aspergillosis *Candida* infections	Fungal cytochrome P450 enzyme inhibitor
Posaconazole (**57**)	Noxafil	Merck	2014	171228-49-2	Invasive aspergillosis *Candida* infections	Sterol 14α-demethylase inhibitor
Isavuconazole (**58**)	Cresemba	Roche	2015	241479-67-4	Invasive aspergillosis *Amphotericin B* is not suitable for the treatment of *Mucormycosis* in adults	Ergosterol synthesis inhibitor

**Table 5 molecules-28-01762-t005:** New drugs for the treatment of drug-resistant tuberculosis.

Generic Name	Brand Name	Company	Approval	CAS Number	Clinical Indications	Molecular Mechanism
Bedaquiline (**59**)	Sirturo	Janssen Therapeutics	2012	843663-66-1	Multidrug-resistant tuberculosis	Adenosine triphosphate synthase inhibitor
Delamanid (**60**)	Deltyba	Otsuka	2014	681492-22-8	Multidrug-resistant tuberculosis	Cell wall synthesis inhibitor
Pretomanid (**61**)	Dovprela	Mylan	2019	187235-37-6	Multidrug-resistant tuberculosis	Cell wall synthesis inhibitor

## Data Availability

Not applicable.

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
