# Peer review of "A Comprehensive Overview of the Antibiotics Approved in the Last Two Decades: Retrospects and Prospects"

_molecules, 2023, doi:10.3390/molecules28041762_

Round 1

Reviewer 1 Report

The manuscript by Shi et al comprehensively reviewed all the approved antibiotics approved over the past two decades. The detailed contents include a discussion of SARs, MOA, and clinical performance of the chemically diversified antibiotics. The authors also gave an outlook on the future direction of antibiotic development.  Overall, this manuscript is well-written and organized and should be of interest to researchers working in the related field.  Therefore, this reviewer recommends it for publication in Molecules pending minor revisions.

1. There are some typos throughout the manuscript. Please proofread carefully and made necessary revisions. Some errors were listed below. In the chemical structure of 6, the acid should be MeSO3H. 

Line 132, CH3O-N=, "3" should be in subscript form.

If some figures from the manuscript were directly copied from previously published work, permission should be obtained and clearly stated in the manuscript.

There are some errors spotted in the reference sections. Page numbers are missing in several places, and the format is not constant.

Author Response

Response to Reviewer 1 Comments

Point 1: There are some typos throughout the manuscript. Please proofread carefully and made necessary revisions. Some errors were listed below. In the chemical structure of 6, the acid should be MeSO3H. Line 132, CH3O-N=, "3" should be in subscript form.

Response 1: Thank you for all your help and advice. They have important guiding significance for my thesis writing! We thank the reviewer for making us note this mistake and we have now made appropriate changes. we apologize for those mistakes. The structural formula of compound 6 has been modified. In line 132, the "3" in "CH3O-N=" has also been changed to subscript form. Apart from the mistakes pointed out by the reviewer above, we have carefully examined the manuscript. Now we indent the first line of each paragraph because we missed line 801. In addition, we improved line breaking in line 835.

Point 2: If some figures from the manuscript were directly copied from previously published work, permission should be obtained and clearly stated in the manuscript.

Response 2: Thanks for the careful reminding of the reviewer, there is no direct copying or pasting of relevant paragraphs or diagrams of other papers in our manuscript, so there is no corresponding permission and explanation.

Point 3: There are some errors spotted in the reference sections. Page numbers are missing in several places, and the format is not constant.

Response 3: Thank you very much for your valuable reminder. Thank the reviewer for pointing out some mistakes in our reference part. We have carefully checked and completed the modification. This work included revising the format of the references (references 3, 16, 27, 72, 75, 114, 116, 122, 123, 125, 133, 142, 154, 155, 164, 165, and 209) and supplementing the page numbers (references 7, 52, 63, 69, 70, 85, 90, 99, 100, 117, 135, 143, 150, 162, 163, 166, 208, 214 and 215), please refer to the manuscript.

We sincerely thank you for all your help with our manuscript.

Reviewer 2 Report

Unfortunately, the paper is very poorly written, particularly whenever anything clinical is mentioned and it included many unscientific information!

I will mention one example which is the clinical indications that were poorly written and arbitrary. For example:

-          Liver diseases (too broad)

-          Sometimes, you mention the bacteria type (gram-positive or negative) and sometimes you do not.

-          There is nothing called sepsis infection. Please delete

-          Please change pneumonia/impetigo/cystitis infection to pneumonia/impetigo/cystitis (remove “infection”)

-          You cannot mix pneumonia and respiratory infections because pneumonia is one of these infections

-          Tigecycline for leukemia! This is a type of cancer and there is no way that an antibiotic would be used for it.

-          Missing major approved indications. For example, skin and soft tissue infections with linezolid, pneumonia with ceftaroline and many others

-          Ventilator-associated tuberculosis! TB is not associated with a ventilator and there is no such terminology.

Even some simple parts like molecular mechanisms included wrong information. E.g., ertapenem being a β-lactamase inhibitor! It is not and it is just one of the beta-lactams. Or meropenem/vaborbactam being a bacterial DNA gyrase and topoisomerase inhibitor! This applies to fluoroquinolones, not beta-lactams at all.

Author Response

Response to Reviewer 2 Comments

Point 1: Unfortunately, the paper is very poorly written, particularly whenever anything clinical is mentioned and it included many unscientific information! I will mention one example which is the clinical indications that were poorly written and arbitrary. For example: Liver diseases (too broad).

Response 1: Thank you for your advice, all your suggestions are very important, they have important guiding significance for my thesis writing and scientific research work! The Liver diseases you mention (too broad) correspond in the manuscript to clinical indications of Alalevonadifloxacin mesylate. Liver diseases is a term we collated from a Thomson Reuters database, please forgive our carelessness. After careful review of the relevant literature about this drug, we realize that Alalevonadifloxacin mesylate is a broad-spectrum fluoroquinolone antibiotic developed by Wockhardt and it was launched in India for the oral treatment of acute bacterial skin and skin structure infections, including diabetic foot infections and concurrent bacteremia caused by methicillin resistant Staphylococcus aureus. Therefore, we choose to delete the liver diseases mentioned before. Thank you again for pointing out our mistake.

Point 2: Sometimes, you mention the bacteria type (gram-positive or negative) and sometimes you do not.

Response 2: Thank you for your comment. This point is mainly reflected in paragraphs and tables in our manuscript. Due to the long time range and too many drugs covered, we considered that if each drug was described in detail from every aspect, the final manuscript would be too long. Therefore, we put different emphasis on the description of each drug in each paragraph, which was reflected in the aspects of structure, pharmacology, efficacy and so on.

Point 3: There is nothing called sepsis infection. Please delete.

Response 3: Thank you very much for your careful review. We have deleted this wrong expression, thank you for your suggestion.

Point 4: Please change pneumonia/impetigo/cystitis infection to pneumonia/impetigo/cystitis (remove “infection”).

Response 4: Thank you for your valuable advice and help. We have corrected this misrepresentation. We have changed pneumonia/pustule/cystitis infection to pneumonia/pustule/cystitis.

Point 5: You cannot mix pneumonia and respiratory infections because pneumonia is one of these infections.

Response 5: Thank you for your suggestions and help with our work. Statements about respiratory infections are included in the manuscript as indications for such drugs as Prulifloxacin (1), Garenoxacin mesylate (7), Lascufloxacin hydrochloride (13), and Telithromycin (33).

Prulifloxacin (1) is a novel fluoroquinolone antibiotic that was launched pursuant to a collaboration between Meiji Seika and Nippon Shinyaku for the oral treatment of systemic bacterial infections, including acute upper respiratory tract infection, bacterial pneumonia, prostatitis, cholecystitis, bacterial enteritis, internal genital infections, otitis media, sinusitis and others. It is currently marketed in a tablet formulation. So the expression for this indication is respiratory infection.

Garenoxacin mesylate (7) is a quinolone antibiotic launched in Japan in 2007 by Toyama (now FUJIFILM Toyama Chemical) for the oral treatment of respiratory and otolaryngologic infections. Regulatory applications have been filed for approval in Europe by Schering-Plough (now Merck & Co.) for the same indications. In 2007, Schering-Plough withdrew its MAA, however, the company plans to resubmit the application with additional data. So the expression for this indication is respiratory infection.

Lascufloxacin hydrochloride (13) is a quinolone synthetic antibacterial developed at Kyorin for the treatment of bacterial infections. In 2020, the oral formulation of lascufloxacin was launched in Japan for the treatment of pharyngitis, stomatitis, tonsillitis (including peritonsillitis and peritonsillar abscess), acute bronchitis, pneumonia, secondary infection of chronic respiratory disease, middle ear infection, and sinusitis, caused by Staphylococcus, Streptococcus, Pneumococcus, Moraxella (Branhamella) Catarrhalis, Klebsiella, Enterobacter, Haemophilus Influenzae, Legionella Pneumophila,  Prevotella and Mycoplasma Pneumoniae, after the approval in 2019. The injectable formulation was approved in Japan in late 2020. So for this indication we've removed the word "respiratory infection".

Telithromycin (33) is a ketolide, a novel form of macrolide antibiotic that is recommended for treatment of community acquired pneumonia. So for this indication we've changed the word "respiratory infection" to "pneumonia".

Point 6: Tigecycline for leukemia! This is a type of cancer and there is no way that an antibiotic would be used for it.

Response 6: Thank you for pointing out these mistakes. We apologize for these careless mistakes and have corrected them.

Point 7: Missing major approved indications. For example, skin and soft tissue infections with linezolid, pneumonia with ceftaroline and many others.

Response 7: We agree with this important point and we thank the reviewers for pointing out our error. We have added key information that the main approved indications for linezolid are skin and soft tissue infections, and the main indications for ceftazoline are pneumonia.

Point 8: Ventilator-associated tuberculosis! TB is not associated with a ventilator and there is no such terminology.

Response 8: We apologize for the confusion and thank the reviewer very much to point out this error. We have removed the incorrect expression about "Ventilator-associated tuberculosis".

Point 9: Like molecular mechanisms included wrong information. E.g., ertapenem being a β-lactamase inhibitor! It is not and it is just one of the beta-lactams. Or meropenem/vaborbactam being a bacterial DNA gyrase and topoisomerase inhibitor! This applies to fluoroquinolones, not beta-lactams at all.

Response 9: We thank the reviewer for pointing out this detail that we described incorrectly. We have removed the incorrect statement that ertapenem is a beta-lactamase inhibitor, and we have also changed the molecular mechanism of meropenem/vaborbactam from "Bacterial DNA gyrase and topoisomerase â…£ inhibitor" to "β-lactamase inhibitor".

Thank you again for all your suggestions and we hope to learn more from you.

Reviewer 3 Report

Dear Authors, I reviewed the manuscript (molecules-2177462) entitled A comprehensive overview of the antibiotics approved in the last two decades: retrospects and prospects. This manuscript presents relevant information about the overuse of antibiotics and their relation with bacterial resistance, presenting state-of-the-art appropriate for the scope of this journal. Besides, the findings obtained in this research are well described and compared with bibliographical references and justify the importance of this obtained data. For this reason, I considered that this manuscript could be accepted for publication in this journal.

Author Response

Response to Reviewer 3 Comments

Point: Dear Authors, I reviewed the manuscript (molecules-2177462) entitled A comprehensive overview of the antibiotics approved in the last two decades: retrospects and prospects. This manuscript presents relevant information about the overuse of antibiotics and their relation with bacterial resistance, presenting state-of-the-art appropriate for the scope of this journal. Besides, the findings obtained in this research are well described and compared with bibliographical references and justify the importance of this obtained data. For this reason, I considered that this manuscript could be accepted for publication in this journal.

Response: Thank you for your recognition of our work and for giving us the opportunity to consider our manuscript for publication in this journal.

We sincerely thank you for your help with our manuscript.